# PqsE Is Essential for RhlR-Dependent Quorum Sensing Regulation in *Pseudomonas aeruginosa*

Marie-Christine Groleau,[a] Thays de Oliveira Pereira,[a] Valérie Dekimpe,[a] Eric Déziel[a]

aCentre Armand-Frappier Santé Biotechnologie, Institut National de la Recherche Scientifique (INRS), Laval, Quebec, Canada

**ABSTRACT** The bacterium *Pseudomonas aeruginosa* has emerged as a central threat in health care settings and can cause a large variety of infections. It expresses an arsenal of virulence factors and a diversity of survival functions, many of which are finely and tightly regulated by an intricate circuitry of three quorum sensing (QS) systems. The *las* system is considered at the top of the QS hierarchy and activates the *rhl* and *pqs* systems. It is composed of the LasR transcriptional regulator and the LasI autoinducer synthase, which produces 3-oxo-$C_{12}$-homoserine lactone (3-oxo-$C_{12}$-HSL), the ligand of LasR. RhlR is the transcriptional regulator for the *rhl* system and is associated with RhlI, which produces its cognate autoinducer $C_4$-HSL. The third QS system is composed of the *pqsABCDE* operon and the MvfR (PqsR) regulator. Pqs-ABCD synthetize 4-hydroxy-2-alkylquinolines (HAQs), which include ligands activating MvfR. PqsE is not required for HAQ production and instead is associated with the expression of genes controlled by the *rhl* system. While RhlR is often considered the main regulator of *rhlI*, we confirmed that LasR is in fact the principal regulator of $C_4$-HSL production and that RhlR regulates *rhlI* and production of $C_4$-HSL essentially only in the absence of LasR by using liquid chromatography-mass spectrometry quantifications and gene expression reporters. Investigating the expression of RhlR targets also clarified that activation of RhlR-dependent QS relies on PqsE, especially when LasR is not functional. This work positions RhlR as the key QS regulator and points to PqsE as an essential effector for full activation of this regulation.

**IMPORTANCE** *Pseudomonas aeruginosa* is a versatile bacterium found in various environments. It can cause severe infections in immunocompromised patients and naturally resists many antibiotics. The World Health Organization listed it among the top priority pathogens for research and development of new antimicrobial compounds. Quorum sensing (QS) is a cell-cell communication mechanism, which is important for *P. aeruginosa* adaptation and pathogenesis. Here, we validate the central role of the PqsE protein in QS particularly by its impact on the regulator RhlR. This study challenges the traditional dogmas of QS regulation in *P. aeruginosa* and ties loose ends in our understanding of the traditional QS circuit by confirming RhlR to be the main QS regulator in *P. aeruginosa*. PqsE could represent an ideal target for the development of new control methods against the virulence of *P. aeruginosa*. This is especially important when considering that LasR-defective mutants frequently arise, e.g., in chronic infections.

**KEYWORDS** cell-cell communication, gene regulation, pyocyanin, virulence factors

Address correspondence to Eric Déziel, eric.deziel@iaf.inrs.ca.

This work is dedicated to the memory of Benjamin Folch (1980 to 2020).

*P*seudomonas aeruginosa*, a bacterium found in a large variety of environments, is most closely associated with human activities (1). This opportunistic human pathogen can cause infections in diverse animals and plants. Its ability to adapt to various conditions has been linked to the many layers of regulation allowing it to control the expression of virulence factors and optimize survival. Quorum sensing (QS) is a mechanism that relies on the release of small signaling molecules as a way to regulate the

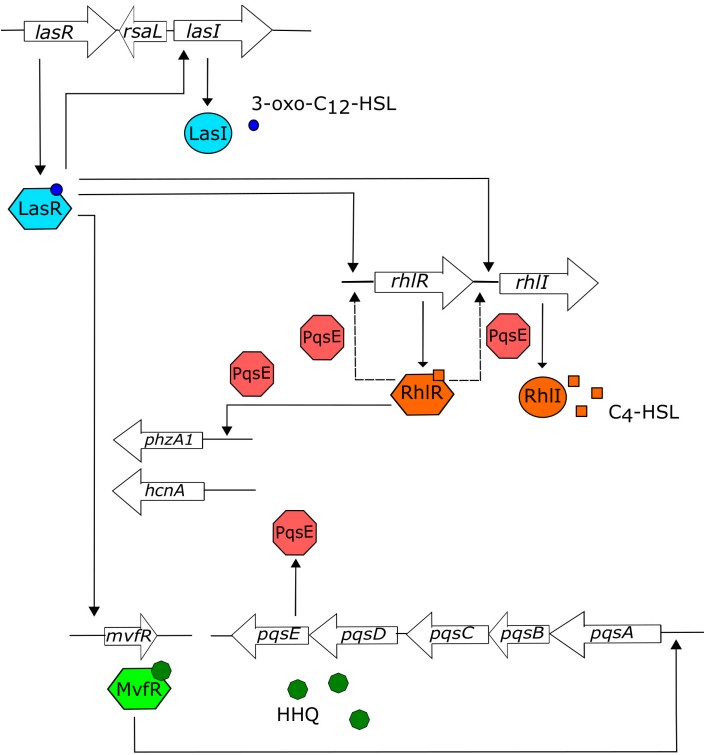

**FIG 1** Schematic representation of quorum sensing regulation by RhlR and PqsE in *Pseudomonas aeruginosa*. The dotted lines represent interactions mostly visible in a LasR-deficient background.

expression of several genes in a population density-dependent manner. In *P. aeruginosa*, three QS systems are hierarchically organized (Fig. 1). The *las* system, which is composed of the transcriptional regulator LasR and the acyl-homoserine lactone (AHL) synthase LasI, is generally considered to be at the top of the regulatory hierarchy. LasR is activated by 3-oxo-$C_{12}$-homoserine lactone (3-oxo-$C_{12}$-HSL), the autoinducing signal produced by LasI. This system regulates several virulence functions such as elastase (LasB) and phospholipase C (PlcB) but also the gene encoding the LasI synthase (2–6). LasR also activates the transcription of the *rhlI* and *rhlR* genes, which code for the AHL synthase RhlI and the transcriptional regulator RhlR (5, 7). In this second AHL-mediated QS system of *P. aeruginosa*, RhlR associates with $C_4$-HSL, produced by RhlI, and activates the transcription of genes implicated in several functions, such as the biosynthesis of rhamnolipids (*rhlAB*), hydrogen cyanide (*hcnABC*), and phenazines (two orthologous *phzABCDEFG* operons) as well as genes encoding lectins (*lecA* and *lecB*) (2, 5, 8–13). The third QS system relies on signaling molecules of the 4-hydroxy-2-alkylquinoline (HAQ) family. The transcriptional regulator MvfR (PqsR) responds to dual ligands 4-hydroxy-2-heptylquinoline (HHQ) and with higher affinity to the *Pseudomonas* quinolone signal (PQS; 3,4-dihydroxy-2-alkylquinoline) to activate the transcription of the *pqsABCDE* operon, which is responsible for their synthesis (14). While LasR activates the transcription of the *mvfR* gene and the *pqs* operon, RhlR has a negative effect on the transcription of *pqsABCDE* (15–17).

LasR-defective mutants frequently arise in various environments (18–22). It could be expected that these mutants would be unable to regulate QS-dependent genes; however, we have shown that RhlR is also able to activate the transcription of LasR target genes when the latter is nonfunctional (23). Indeed, LasR-defective strains expressing RhlR-regulated functions are found (22, 24, 25), implying that QS is not abolished in the absence of LasR. In recent work, a *lasR* mutant isolated from the lungs of an individual with cystic fibrosis expressed a *rhl* system that acted independently of the *las* system (26). It allowed this strain to produce factors essential for its growth

under a specific condition that would normally require a functional LasR. When evolved under controlled conditions, this strain gained a mutation in MvfR (PqsR) making it unable to produce PQS and to activate the RhlR-dependent genes, highlighting the link between the *pqs* operon and RhlR.

Although a thioesterase activity of PqsE could participate in the biosynthesis of HAQs (27), the protein encoded by the last gene of the *pqs* operon is not required, since a *pqsE* mutant shows no defect in HAQ production (14). On the other hand, PqsE is implicated in the regulation of genes that include many of the RhlR-dependent targets, such as the *phz* and *hcn* operons and the *lecA* gene, through an unknown mechanism (28–33). An impact of PqsE on the RhlR-dependent regulon was proposed; for instance, PqsE could enhance the affinity of RhlR for $C_4$-HSL (28) or even synthesize an alternative ligand for RhlR (34). Importantly, such function is independent of its thioesterase function, as inhibitors of this activity had no impact on the regulatory functions of PqsE (27, 28).

In this study, we validate that activation of RhlR-dependent QS strongly relies on the presence of a functional PqsE and reveal that this is especially important for activation of the *rhl* system in cases where LasR is not functional. This makes RhlR the key QS regulator and points to PqsE as an essential effector for full activation of this regulation. These findings thus strengthen the position of RhlR as the master regulator of QS and place PqsE at the center of QS regulatory circuitry in *P. aeruginosa*.

## RESULTS AND DISCUSSION

**RhlR is not the main activator of $C_4$-HSL production.** Quorum sensing regulation is typically described as a partnership between a LuxI-type AHL synthase and a LuxR-type transcriptional regulator. The LuxR-type regulator is activated by a cognate AHL and then regulates the transcription of target genes as well as the gene encoding the synthase, which upregulates AHL production, resulting in an autoinducing loop. In *P. aeruginosa*, the 3-oxo-$C_{12}$-HSL synthase LasI is associated with the LasR regulator and the $C_4$-HSL synthase RhlI with the RhlR regulator. Interestingly, LasR regulates the transcription of both *rhlI* and *rhlR* genes (2, 5, 7, 35); actually, it has been argued that LasR, and not RhlR, is the primary regulator of *rhlI* (35). Accordingly, we previously reported that $C_4$-HSL production is decreased in a *lasR* mutant (23, 26). Indeed, a study in strain 148 showed that LasR binds the *lux* box found in the promoter region of *rhlI* but that RhlR does not (36), while other studies showing a direct regulation of *rhlI* by RhlR were actually performed in a heterologous host, in the absence of LasR (7, 35). Together, these reports would suggest that RhlR mostly activates the transcription of *rhlI* when LasR is unable to.

To verify that RhlR is not the main regulator of $C_4$-HSL production in a LasR-positive background, we measured concentrations of this AHL in cultures using liquid chromatography-tandem mass spectrometry (LC-MS/MS). The production of $C_4$-HSL is only detectable at the stationary phase in a *lasR* mutant, while in a *rhlR* mutant, the production is only slightly delayed compared to that of wild-type (WT) *P. aeruginosa* strain PA14 (Fig. 2). This concurs with the often-overlooked idea (e.g. see reference 37) that it is LasR, rather than RhlR, that is primarily responsible for activating the transcription of *rhlI* and thus the production of $C_4$-HSL, the ligand of RhlR. Interestingly, production is even more diminished in a double *lasR pqsE* mutant, while it is not affected at all in the Δ*pqsE* mutant, indicating PqsE has a role in LasR-independent activation of $C_4$-HSL production (Fig. 2).

**PqsE is important for LasR-independent quorum sensing.** A plausible explanation for the results presented in Fig. 2 is that RhlR is a secondary regulator of *rhlI*, mostly important in the absence of LasR only, and that the absence of PqsE negatively affects the activity of RhlR only when LasR is not functional. To verify this hypothesis, we needed to investigate the activity of RhlR through one of its primary targets. Phenazines are redox-active metabolites produced by *P. aeruginosa* and are synthetized via two redundant operons: *phzA1-G1* (*phz1*) and *phzA2-G2* (*phz2*). These operons are almost identical and encode proteins that catalyze the synthesis of phenazine-1-

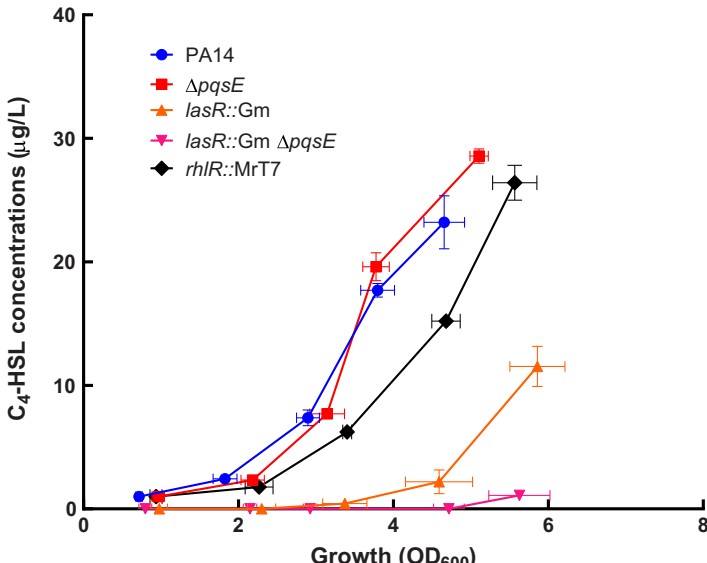

**FIG 2** $C_4$-HSL production depends mostly on LasR. $C_4$-HSL production was measured in cultures of PA14 and Δ*pqsE*, *lasR*::Gm, *lasR*::Gm Δ*pqsE*, and *rhlR*::MrT7 mutants at different time points during growth. The values are means ± standard deviations (error bars) from three replicates.

carboxylic acid (PCA). PCA converts into derivatives such as pyocyanin, the blue pigment characteristic of *P. aeruginosa* cultures (38). The *phz* operons are differentially regulated depending on conditions, but the *phz1* operon shows higher expression than *phz2* in planktonic cultures of strain PA14 (39). The promoter of the *phz1* operon contains a *las* box which can be recognized by both LasR and RhlR (40). We measured the activity of a chromosomal *phzA1-lux* reporter in both *lasR* and *rhlR* mutants to verify their involvement in the regulation of the transcription of the *phz1* operon (Fig. 3). The transcription of *phz1* is completely abolished in a *rhlR* mutant but it is still observed in a *lasR* mutant, although it starts much later than for the WT (after an optical density at 600 nm [$OD_{600}$] of 4.0). This is consistent with the delayed production of pyocyanin (23, 41) and $C_4$-HSL (Fig. 2) observed in cultures of a *lasR* mutant. Since we know that

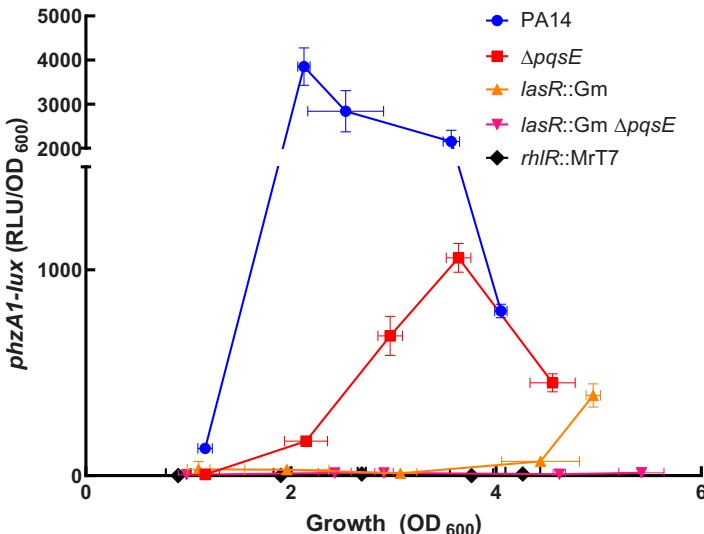

**FIG 3** Transcription of the *phz1* operon absolutely requires RhlR and PqsE in a *lasR*-negative background. Luminescence of a *phzA1-lux* chromosomal reporter was measured in *P. aeruginosa* PA14 and various isogenic mutants at different time points during growth. The values are means ± standard deviations (error bars) from three replicates.

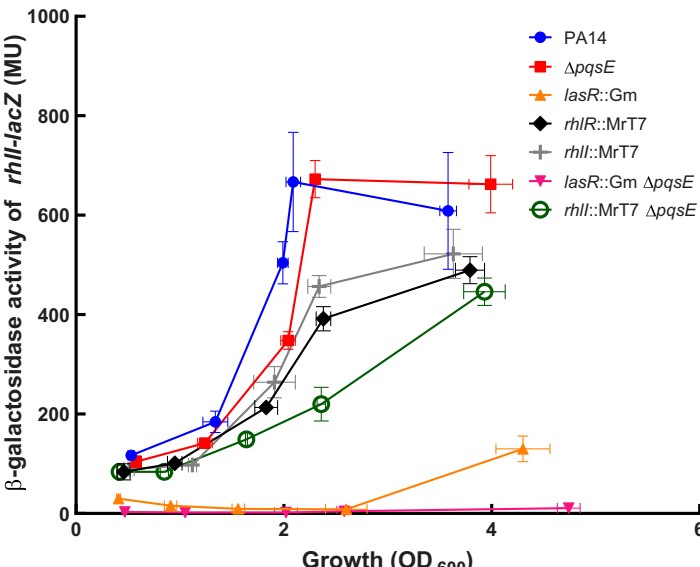

**FIG 4** The transcription of *rhlI* requires PqsE in a *lasR* mutant. The β-galactosidase activity of a *rhlI-lacZ* reporter was measured in various backgrounds at different time points during growth. The values are means ± standard deviations (error bars) from three replicates.

transcription of *phz1* and production of pyocyanin are abrogated in a double *lasR rhlR* mutant (23, 41), these results indicate that RhlR, but not LasR, regulates the transcription of *phzA1* and that RhlR is responsible for the late activation of *phzA1* expression in a *lasR*-negative background. We used transcription of the *phz1* operon to further study the influence of PqsE on RhlR-dependent regulation. Even if cultures of a *pqsE* mutant do not show any visible pyocyanin, we still observe clear expression of *phz1* (Fig. 3). Since there is no pyocyanin produced in the WT until an $OD_{600}$ of around 2.5 even if there is expression from the *phzA1* promoter, there seems to be a minimal level of expression of *phz* genes for detectable pyocyanin. Also, pyocyanin is not a direct product of the *phz* operons and it is possible that other enzymes (e.g., PhzM or PhzS) implicated in the conversion of PCA to pyocyanin do not follow the same pattern of expression in this background (29). The transcription of *phzA1* is completely abolished in a double *lasR pqsE* mutant. Many studies report an impact of PQS-dependent QS on the regulation of the *phz* operons or pyocyanin production (28, 31, 39, 41, 42). More specifically, this effect necessitates a functional PqsE (28, 42).

Because LasR regulates the expression of *rhlI* (5, 7, 23), we performed a β-galactosidase assay using a *rhlI-lacZ* reporter to verify the impact of PqsE on the transcription of *rhlI*. As expected, transcription of *rhlI* is much delayed in a *lasR* mutant (Fig. 4). This is compatible with the late activation of *phz1* we observed (Fig. 3) and is apparently occurring because RhlR takes the relay in activating the transcription of *rhlI* following the initial activation by LasR. When the *pqsE* gene is inactivated in a *lasR* background, very low transcription of *rhlI* is observed (Fig. 4) which concurs with the production of $C_4$-HSL in this background (Fig. 2) and which agrees with a PqsE-dependent activity of RhlR. Again, since RhlR takes over regulating the production of $C_4$-HSL following the initial activation by LasR, the transcription of *rhlI* slows down in *rhlR* and *rhlI* mutants after an $OD_{600}$ of 2.0, when LasR main activity is decreasing (the levels of 3-oxo-$C_{12}$-HSL are rapidly declining) (23, 31). Together, these data point to a role for PqsE in LasR-independent regulation of the *rhl* system.

**PqsE/RhlR/$C_4$-HSL collude to activate LasR-independent quorum sensing.** Since $C_4$-HSL has an effect on RhlR activity (2, 7, 28), we needed to better understand the functional complementary of $C_4$-HSL with PqsE in modulating the activity of RhlR. We measured the activity of the *phzA1-lux* reporter in a *rhlI* mutant as well as in a double *rhlI pqsE* mutant. Transcription of *phzA1* in the *rhlI* mutant was delayed, but not

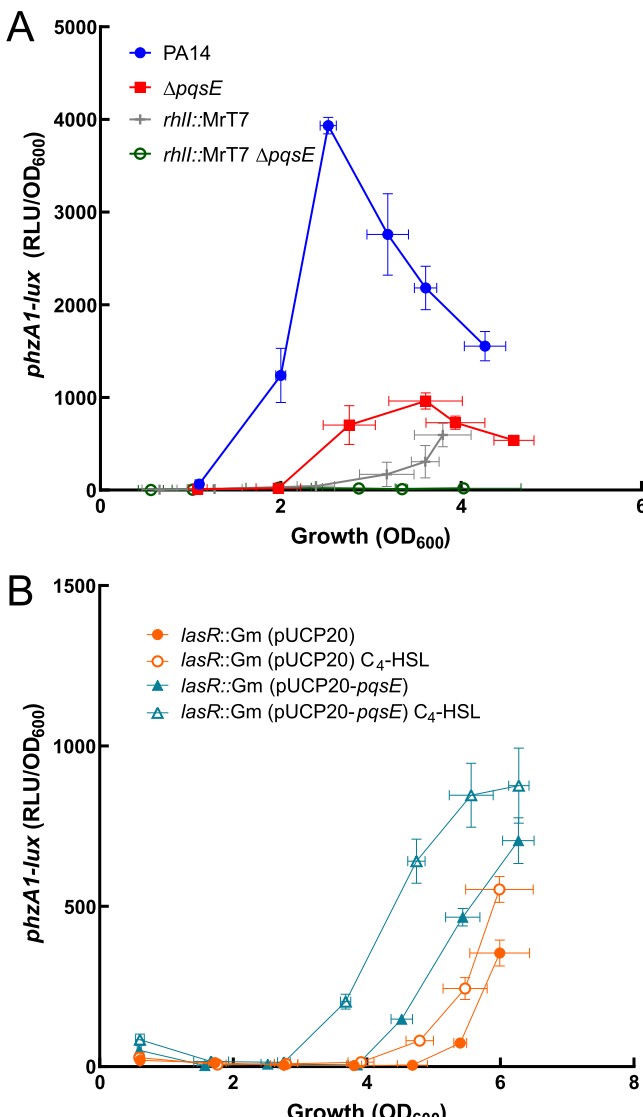

**FIG 5** The impacts of C$_4$-HSL and PqsE on RhlR activity. The expression of *phzA1-lux* is cumulative. (A) Luminescence of a *phzA1-lux* chromosomal reporter was measured in WT and isogenic Δ*pqsE* and *rhlI*::MrT7 mutants and double mutant *rhlI*::MrT7 Δ*pqsE* at different time points during growth. (B) Luminescence of the *phzA1-lux* chromosomal reporter was measured in a *lasR*::Gm background with either empty vector pUCP20 or pUCP20-*pqsE* with or without the addition of C$_4$-HSL. The values are means ± standard deviations (error bars) from three replicates.

abolished, suggesting that RhlR utilizes its AHL ligand to activate the *phz1* operon but that its presence is not essential (Fig. 5A). However, when both C$_4$-HSL and PqsE are absent (*rhlI pqsE* double-negative background), there is no residual transcription of *phz1* (Fig. 5A), like in the *rhlR*-negative background (Fig. 3). The profile of expression of *phz1* significantly differs between *pqsE* and *rhlI* mutants (P values of <0.05 from OD$_{600}$s of 3.0 to 3.6). In the *pqsE* mutant, the expression starts at an OD$_{600}$ of around 2.0, while in the *rhlI* mutant, it starts later (OD$_{600}$ of around 3.5) and keeps augmenting through the rest of the growth curve. This suggests that both elements increase the activity of RhlR through different mechanisms.

Since the absence of LasR seems to impose the requirement for PqsE to achieve efficient RhlR activity, we overexpressed *pqsE* in a *lasR*-null background. As previously shown (43), the constitutive expression of PqsE augments and advances the transcription of *phzA1* (Fig. 5B). When we added exogenous C$_4$-HSL in the *lasR* mutant bearing a plasmid-borne *pqsE*, the transcription of *phz1* started even earlier and reached higher

levels than with either one separately (*P* values of 0.046 and 0.002, respectively). Farrow et al. (28) proposed that PqsE acts by enhancing the affinity of RhlR for $C_4$-HSL. However, we see that PqsE increases the activity of RhlR even in the absence of RhlI (Fig. 4 and 5A), thus not supporting this hypothesis; our data suggest that RhlR full activity depends on both $C_4$-HSL and PqsE and that their impact is cumulative.

The induction of RhlR activity by PqsE in the absence of *rhlI* could be explained by the proposed PqsE-dependent production of a putative alternative RhlR ligand. Indeed, Mujurkhee and colleagues (13) observed activation of *rhlA* transcription by adding culture-free fluids from a Δ*rhlI* mutant to a QS mutant expressing *rhlR* under the control of an arabinose-inducible promoter. They proposed in a subsequent study that this activity was PqsE dependent (34). We thus tested the effect of *pqsE*, *rhlI*, and *rhlI pqsE* mutants cell-free culture fluids on the activation of *phzA1-lux* in the *rhlI pqsE* double-negative background. As expected, the activity of the reporter is strongly induced by culture supernatants from PA14 or a *pqsE* mutant (which both contain $C_4$-HSL). On the other hand, there is no activation by supernatants from *rhlI* and *rhlI pqsE* mutants (see Fig. S1 in the supplemental material), even when combined with an overexpression of *rhlR* (data not shown). This argues against an unknown RhlR inducer whose production would require PqsE. The same results were obtained when using an *hcnA-lacZ* reporter (data not shown).

To validate our model, we looked at the regulation of the *hcnABC* operon, a dual target of both LasR and RhlR (12, 41), and obtained results similar to what we observed for the *phz1* operon and the *rhlI* gene (see Fig. S2). Taken altogether, our data highlight a possible homeostatic loop between RhlR-RhlI-PqsE and demonstrate that PqsE is essential for maintaining control of RhlR-dependent QS functions in a LasR-independent way.

**Excess RhlR, but not $C_4$-HSL, can overcome a PqsE deficiency.** We then sought to better understand how $C_4$-HSL and PqsE both contribute to RhlR activity. First, we verified if overproduction of $C_4$-HSL could counterbalance a lack of PqsE. It was already shown that adding $C_4$-HSL alone could not restore pyocyanin production in a triple Δ*lasR* Δ*rhlI* Δ*pqsA* mutant, but that adding PQS and $C_4$-HSL together could (41). We thus used a plasmid-borne p*lac-rhlI* for constitutive $C_4$-HSL production and measured its effects on the transcription of *phz1* and on pyocyanin production in various back-grounds. Overexpression of *rhlI* complements the transcription of *phz1* in a *lasR* mutant enough to show pyocyanin production at the stationary phase (Fig. 6A; see also Fig. S3). As expected, this complementation was not as efficient when a *pqsE* mutation was added to the *lasR*-negative background, as there was even less transcription of *phz1* (*P* values of <0.05 at all growth phases) (Fig. 6A). Taken together, these results confirm that $C_4$-HSL cannot counterbalance the absence of PqsE and highlight an important role for PqsE in regulating RhlR-dependent genes; this is especially striking in the absence of LasR.

We then looked at the overexpression of RhlR, since it partially restores pyocyanin production in a Δ*pqsE* background (30). We observed an augmentation in both the transcription of *phzA1* and pyocyanin production (Fig. 6B and S3). Figure S3 shows that when RhlR is overexpressed, both *lasR* and *lasR pqsE* mutants produce higher levels of pyocyanin, coupled with strong activation of *phzA1-lux* expression in both back-grounds. This is the first ever report of restoration of *phz1* transcription and pyocyanin production in the absence of PqsE. Surprisingly, we observed a discrepancy between the transcription from the *phzA1* promoter and pyocyanin production, which indicates that the transcription of the target genes shows a more realistic portrait of the activity of RhlR than only looking at pyocyanin production.

Further supporting our model, the transcription of *phzA1* and the production of pyocyanin when *rhlR* was overexpressed were higher in the *lasR* mutant than in the *lasR pqsE* mutant (*P* value of <0.05 at $OD_{600}$s of 2.0 to 4.0), and these results again confirm an effect of PqsE on RhlR activity.

**PqsE affects RhlR regulatory activity on its targets, including itself, in the absence of LasR.** The very late activity of *phz1* in *lasR*-negative backgrounds can be

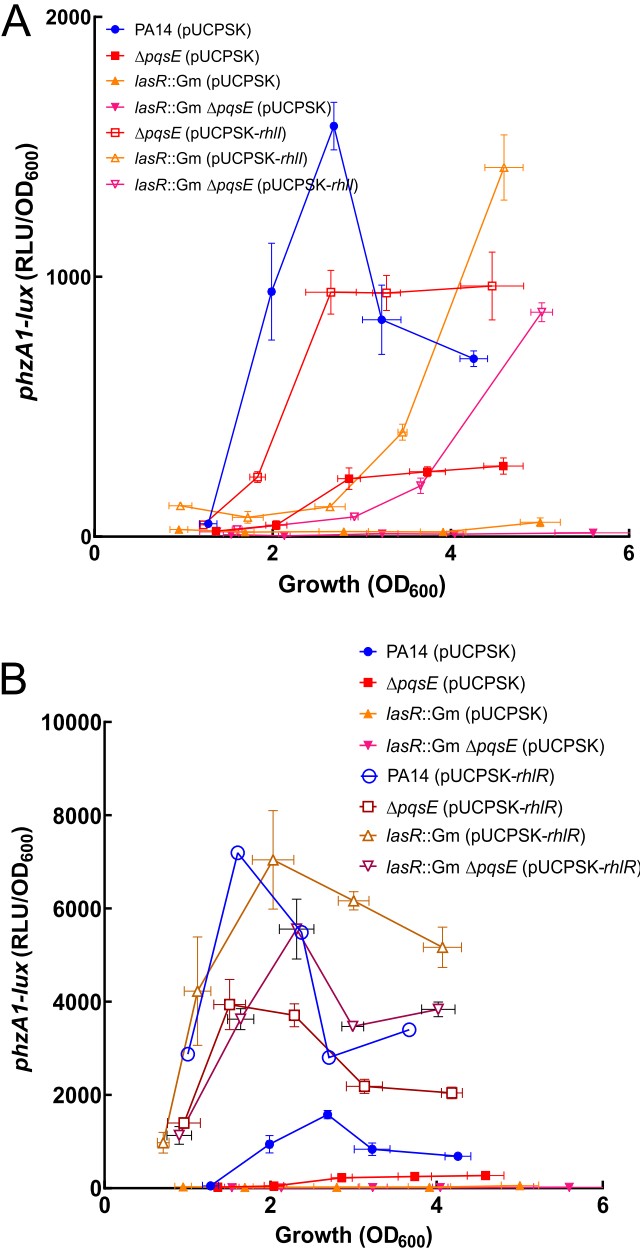

**FIG 6** Effects of *rhlI* and *rhlR* overexpression on *phz1* transcription. Luminescence of a *phzA1-lux* chromosomal reporter was measured in PA14, Δ*pqsE*, *lasR*::Gm, and *lasR*::Gm Δ*pqsE* mutants at different time points during growth with overexpression of RhlI (A) or RhlR (B). The values are means ± standard deviations (error bars) from three replicates.

explained by low levels of RhlR, whose initial transcription also requires LasR (2, 5–7, 35). When measuring the activity of an *rhlR-lacZ* reporter, there was indeed a lower transcription of *rhlR* in a *lasR* mutant (Fig. 7). Since overexpression of *rhlI* did not lead to full activation of the *phz* genes in a double *lasR pqsE* mutant background (Fig. 6A), we hypothesized that this was instead caused by low transcription of the *rhlR* gene. Interestingly, the level of *rhlR* transcription was even lower in the double *lasR pqsE* mutant background than in the single *lasR* mutant. This result is unexpected since the transcription of *rhlR* is weakly affected in a *pqsE*-null background (30). Because RhlR can activate the target genes of LasR when the latter is absent (23), we hypothesized that RhlR could therefore regulate itself, explaining the impact of PqsE only in the absence of LasR. Transcription of *rhlR-lacZ* was accordingly lower in a double *lasR rhlR* mutant,

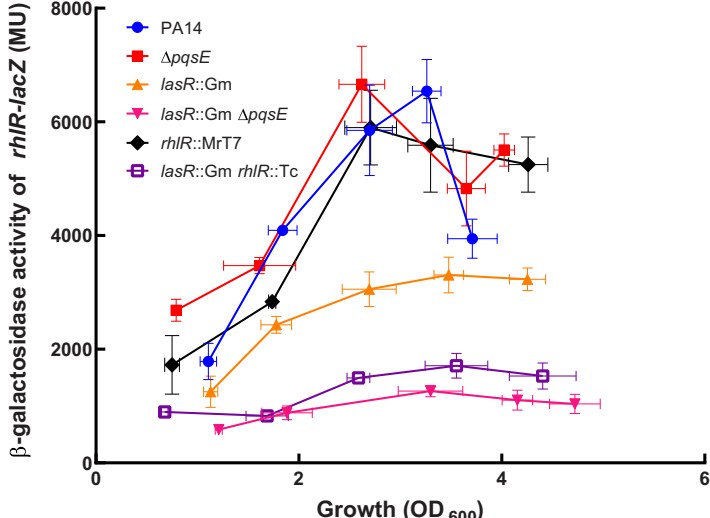

**FIG 7** PqsE affects RhlR autoregulation. The β-galactosidase activity of a *rhlR-lacZ* reporter was measured in various backgrounds at different time points during growth. The values are means ± standard deviations (error bars) frrom three replicates.

to levels similar to those in the *lasR pqsE* mutant (nonsignificant, $P > 0.05$ at all growth phases) (Fig. 7). This indicates that RhlR directs its own transcription only in the absence of LasR and that PqsE is important for this activity. These data confirm that PqsE is an essential element in RhlR activity when LasR is not functional.

**Conclusion.** The complex quorum sensing circuitry of *P. aeruginosa* has been extensively studied, and we know all three systems are intimately intertwined (44, 45). Although RhlR is often believed to form a traditional autoinducing pair with *rhlI*, we confirm here that LasR really is the main activator of $C_4$-HSL production and that RhlR activation of *rhlI* is mainly observed in the absence of a functional LasR. LasR is also an activator of the *pqs* operon and thus of PqsE. However, production of $C_4$-HSL and PQS are not completely abolished in a *lasR* mutant, only delayed. In a *lasR*-null background, the importance of RhlR and PqsE on the activation of *phzA1*, *rhlI*, or *hcnA* is higher than in the WT, since LasR is at the top of the regulation cascade. This allowed us to observe that RhlR is able to fully activate target genes only if PqsE is present. The function of PqsE has been a subject of many studies but is still enigmatic (32). In this work, we show that PqsE most likely promotes the function of RhlR and that this effect seems independent of the presence of $C_4$-HSL or another putative ligand, as previously proposed.

Under laboratory conditions, *P. aeruginosa* can afford a late activation of QS or even no activation of QS at all. In a more competitive environment, it is likely there is pressure to control these genes and to activate their transcription independently of LasR when necessary. PqsE could thus be important as a trigger for stronger and/or earlier RhlR activity. A growing number of studies report on the presence of LasR-deficient variants in chronic infections settings (18, 19, 22). With the absence of a functional LasR in these strains, the traditional QS hierarchy is altered and independent expression of RhlR becomes necessary for the bacteria to activate functions important for survival in hosts, such as virulence factors (like exoproteases and HCN) or biofilm formation (rhamnolipids and lectins).

Importantly, among LasR-deficient *P. aeruginosa* strains isolated from clinical settings, some still express a functional quorum sensing response through the activity of RhlR, independently of LasR (22, 26). Since this study was limited to the prototypical strain PA14, it will be important to extend our findings and investigate the implication of PqsE in the activation of the RhlR regulon in diverse clinical and environmental isolates in order to better understand its role in QS gene regulation in *P. aeruginosa*.

**TABLE 1** Strains used in this study

| Strain | Description | Reference or source |
|---|---|---|
| *E. coli* | | |
| DH5α | F⁻, *φ80dlacZ*ΔM15 Δ(*lacZYA-argF*)U169 *deoR recA1 endA1 hsdR17*(r$_K$⁻ m$_K$⁺) *phoA supE44* λ⁻ *thi-1 gyrA96 relA1* | Lab collection |
| χ7213 | *thr-1 leuB6 fhuA21 lacY1 glnV44 recA1 ΔasdA4 Δ(zhf-2::Tn10) thi-1 RP4-2-Tc::Mu* [λ *pir*] | Lab collection |
| *P. aeruginosa* | | |
| ED14/PA14 | Clinical isolate UCBPP-PA14 | 50 |
| ED36 | Δ*pqsE* | 14 |
| ED69 | *lasR*::Gm | 14 |
| ED247 | *lasR*::Gm Δ*pqsE* | This study |
| ED503 | *rhlR*::Gm | 30 |
| ED297 | *rhlI*::MrT7 | 51 |
| ED3579 | *rhlI*::MrT7 Δ*pqsE* | This study |
| ED266 | *lasR*::Gm *rhlR*::Tc | 23 |

## MATERIALS AND METHODS

**Strains, plasmids, and growth conditions.** Bacterial strains are listed in Table 1. Plasmids used in this study are listed in Table 2. Unless otherwise stated, bacteria were routinely grown in tryptic soy broth (TSB; BD Difco, Canada) at 37°C in a TC-7 roller drum (NB, Canada) at 240 rpm or on lysogeny broth (LB) agar plates. When antibiotics were needed, the following concentrations were used: for *Escherichia coli*, 15 μg/ml tetracycline and 100 μg/ml carbenicillin, for *P. aeruginosa*, 100 μg/ml gentamicin, tetracycline at 125 μg/ml (solid) or 75 μg/ml (liquid), and 250 μg/ml carbenicillin. Diaminopimelic acid (DAP) was added to cultures of the auxotroph *E. coli* χ7213 at 62.5 μg/ml. All plasmids were transformed in bacteria by electroporation (46).

All experiments presented in this work were performed with three biological replicates and repeated at least twice.

**Construction of the double Δ*pqsE* mutants.** A knockout in both *rhlI* and *pqsE* was constructed by transfer between chromosomes (46). The genomic DNA (gDNA) of strain ED297 *rhlI*::MrT7 was extracted using the EasyPure bacteria genomic kit (Trans Gen Biotech, China). Three milliliters of an overnight culture of Δ*pqsE* was centrifuged (16,000 × *g*, 2 min) in separate microtubes. Pellets were washed twice with 300 mM sucrose. The pellets were combined in a final volume of 100 μl 300 mM sucrose. Five hundred nanograms of gDNA was added to the bacterial suspension, and the mixture was transferred to a 0.2-mm electroporation cuvette. The cells were electroporated at 2,500 V, immediately transferred to 1 ml LB, and incubated at 37°C for 1 h. Selection was performed on LB agar containing gentamicin. Clones were selected and verified by PCR. The *lasR*::Gm mutation was introduced in the Δ*pqsE* background by allelic exchange using pSB219.9A as described (14, 47).

**Construction of *phz1-lux* chromosomal reporter strains.** The mini-CTX-*phz1-lux* construct was integrated into the chromosomes of PA14 WT and mutants by conjugation on LB agar plates containing DAP with *E. coli* χ7213 containing the pCDS101 plasmid. Selection was performed on LB agar plates containing tetracycline.

**β-Galactosidase activity assays and luminescence reporter measurements.** Strains containing the reporter fusions were grown overnight in TSB with appropriate antibiotics and diluted at an OD$_{600}$ of 0.05 in TSB. For *lacZ* reporter assays, culture samples were regularly taken for determination of growth (OD$_{600}$) and β-galactosidase activity (48). For *lux* reporter assays, luminescence was measured using a Cytation 3 multimode microplate reader (BioTek Instruments, USA). When mentioned, C$_4$-HSL was added at a final concentration of 20 μM from a stock solution prepared in high-performance liquid chromatography (HPLC)-grade acetonitrile. Acetonitrile only was added in controls. All OD$_{600}$ measurements were performed with a NanoDrop ND100 spectrophotometer (Thermo Fisher Scientific, Canada).

**TABLE 2** Plasmids used in this study

| Plasmid | Description | Reference or source |
|---|---|---|
| pCDS101 | Promoter of *phz1* in mini-CTX-*lux*, Tet$^r$ | 52 |
| pPCS1002 | *rhlR-lacZ* reporter, Carb$^r$ | 2 |
| pSB219.9A | pRIC380 carrying *lasR*::Gm | 47 |
| pME3846 | *rhlI-lacZ* translational reporter, Tet$^r$ | 53 |
| pME3826 | *hcnA-lacZ* translational reporter, Tet$^r$ | 54 |
| pUCPSK | *Pseudomonas* and *Escherichia* shuttle vector, Carb$^r$ | 55 |
| pMIC62 | *rhlR* gene under control of the *lac* promoter in pUCPSK | John Mattick |
| pUCP*rhlI* | *rhlI* gene under control of the *lac* promoter in pUCPSK | 47 |
| pUCP20 | *Pseudomonas* and *Escherichia* shuttle vector, Carb$^r$ | 56 |
| pUCP20-*pqsE* | *pqsE* gene under control of the *lac* promoter in pUCP20, Carb$^r$ | 57 |

**Pyocyanin quantification.** Overnight cultures of PA14 and mutants were diluted to an $OD_{600}$ of 0.05 in TSB and grown until an $OD_{600}$ of 4 to 5 was reached. Cells were removed by centrifugation at $13,000 \times g$ for 5 min, and the cleared supernatant was transferred to 96-well microplates. The absorbance at 695 nm was measured using a Cytation 3 multimode microplate reader. Pyocyanin production was determined by dividing the $OD_{695}$ by the $OD_{600}$.

**Quantification of AHLs.** Analyses were performed by liquid chromatography-mass spectrometry (LC-MS) as described before with 5,6,7,8-tetradeutero-4-hydroxy-2-heptylquinoline (HHQ-d4) as an internal standard. (49).

**Data analysis.** Statistical analyses were performed using R software version 3.6.3 (http://www.R -project.org) using one-way analysis of variance (ANOVA) with Tukey *post hoc* tests at different stages of growth. All conclusions discussed in this paper were based on significant differences. Probability (*P*) values of less than 0.05 were considered significant.

## SUPPLEMENTAL MATERIAL

Supplemental material is available online only.

**FIG S1**, PDF file, 0.1 MB.

**FIG S2**, PDF file, 0.2 MB.

**FIG S3**, PDF file, 0.1 MB.

## ACKNOWLEDGMENTS

We thank Sylvain Milot and Marianne Piochon for their help with the LC-MS analyses, Philippe Constant for his help with statistical analyses, and Alison Besse and Fabrice Jean-Pierre for critical reading of the manuscript.

This study was supported by Canadian Institutes of Health Research (CIHR) operating grants MOP-97888 and MOP-142466 to E.D.

E.D. holds the Canada Research Chair in Sociomicrobiology. The funders had no role in study design, data collection and interpretation, or the decision to submit the work for publication.

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
