## [Reviewer comments · mSystems]

PqsE is essential for RhIR-dependent quorum sensing regulation in *Pseudomonas aeruginosa*

Marie-Christine Groleau, Thays de Oliveira Pereira, Valérie Dekimpe, and Eric Déziel

Corresponding Author(s): Eric Déziel, INRS-Armand-Frappier Santé Biotechnologie

Review Timeline:

Submission Date:	March 3, 2020
Editorial Decision:	April 1, 2020
Revision Received:	April 14, 2020
Accepted:	May 5, 2020

Editor: Elizabeth Shank

Reviewer(s): Disclosure of reviewer identity is with reference to reviewer comments included in decision letter(s). The following individuals involved in review of your submission have agreed to reveal their identity: Peter Jorth (Reviewer #1)

Transaction Report:

DOI: <https://doi.org/10.1128/mSystems.00194-20>

April 1, 2020

Dr. Eric Déziel
INRS-Armand-Frappier Santé Biotechnologie
531 Boul. des Prairies
LAVAL, QC H7V1B7
Canada

Re: mSystems00194-20 (PqsE is essential for RhIR-dependent quorum sensing regulation in *Pseudomonas aeruginosa*)

Dear Dr. Eric Déziel:

Below you will find the comments of the reviewers. As you will see, your manuscript would require some minor modifications before final consideration for publication, specifically strengthening the statistical analysis of your data and improving the clarity of your abstract, among the other reviewer suggestions below.

To submit your modified manuscript, log onto the eJP submission site at <https://msystems.msubmit.net/cgi-bin/main.plex>. If you cannot remember your password, click the "Can't remember your password?" link and follow the instructions on the screen. Go to Author Tasks and click the appropriate manuscript title to begin the resubmission process. The information that you entered when you first submitted the paper will be displayed. Please update the information as necessary. Provide (1) point-by-point responses to the issues raised by the reviewers as file type "Response to Reviewers," not in your cover letter, and (2) a PDF file that indicates the changes from the original submission (by highlighting or underlining the changes) as file type "Marked Up Manuscript - For Review Only."

Due to the SARS-CoV-2 pandemic, our typical 60 day deadline for revisions will not be applied. I hope that you will be able to submit a revised manuscript soon, but want to reassure you that the journal will be flexible in terms of timing, particularly if experimental revisions are needed. When you are ready to resubmit, please know that our staff and Editors are working remotely and handling submissions without delay. If you do not wish to modify the manuscript and prefer to submit it to another journal, please notify me of your decision immediately so that the manuscript may be formally withdrawn from consideration by mSystems.

To avoid unnecessary delay in publication should your modified manuscript be accepted, it is important that all elements you upload meet the technical requirements for production. I strongly recommend that you check your digital images using the Rapid Inspector tool at <http://rapidinspector.cadmus.com/RapidInspector/zmw/>.

Sincerely,

Elizabeth Shank

Editor, mSystems

Journals Department
Reviewer comments:

Reviewer #1 (Comments for the Author):

In this manuscript the authors revisit the multilayered *Pseudomonas aeruginosa* quorum sensing (QS) circuitry, which includes the *las*, *rhl*, and *pqs* QS systems which involve the 3-oxo-C12-HSL, C4-HSL, and PQS signaling molecules, respectively. In this hierarchy the *las* QS system is first activated during growth followed by the downstream *rhl* and *pqs* systems. Recently, it has also become appreciated that LasR, and not RhlR, is the major driver of C4-HSL production in WT *P. aeruginosa*. However, it is becoming increasingly apparent that the *lasR* regulator is frequently mutated during chronic infections, and QS genes can be still be expressed in *lasR* mutants. Here the authors investigated the role of the C4-HSL regulator RhlR, the C4-HSL synthetase RhlI, and PqsE in modulating QS gene expression in *lasR* mutant backgrounds. Using a series of mutants generated in the PA14 reference strain, the authors measured QS gene expression, QS virulence factors, and C4-HSL signaling molecule production. Ultimately, these experiments show that PqsE enhances RhlR-dependent QS in a *lasR* mutant background.

Overall, this is an interesting paper that makes an important contribution towards our understanding of *lasR* mutant *P. aeruginosa*, which are common mutants found in many chronic infections, including cystic fibrosis. The experiments are thoroughly controlled and employ a number of complementary methods. However, the experimental design lacks statistical tests to compare the different mutants. The authors plot most data as QS gene expression vs culture density. This temporal analysis is valuable, because it shows that some genes are activated at later phases of growth in the different mutant backgrounds. Yet, in its current presentation, the data analyses lack statistical rigor, and these comparisons are not as strong as they might be if a statistical test was included for each experiment. While the data were not captured for all strains at exactly the same optical density, the data could be binned in a way to compare the different mutants at comparable ODs (see major comments below).

No new experiments are needed to back up the conclusions in this paper; however, reanalysis with statistics would strengthen the data.

Major comments

The abstract is somewhat confusing and could be written more clearly. In its current form, the abstract goes from background information to results, to background information and more results. It makes it somewhat challenging to determine what was known before this study, what was done (no methods are mentioned), and what was actually determined. Admittedly, this is a complicated story, but the abstract should be rewritten for clarity.

Figures: The line graphs could be maintained to still show the trends, but the data could be plotted a second way to enable multiple ANOVA tests to compare C4-HSL production and reporter activities at different phases of growth. For example, the data could be binned by OD600. In Figure 2 the bins might be OD600 0-1.5, 1.8-2.2, 2.5-3.5, 3.75-4.0, 4.1-4.5, 5.0-6.0. Within each bin, an ANOVA test could be performed with multiple comparisons to determine if strains are statistically significantly different in the given experimental measurements at the different phases of growth.

Some of the figures (1, 5, and 6) were blurry.

The authors do not discuss any of the limitations to their study. Some limitations should be addressed, including the use of only one strain, PA14; as well as the use of many marked mutants. Why weren't clean deletions used? Have the mutants used in this study been previously complemented?

Reviewer #2 (Comments for the Author):

This paper examines the roles of PqsE and RhIR specifically in the absence of LasR. Because lasR mutants are commonly found in infections, it is important to understand the QS hierarchy in the light of this and this paper sheds further welcome information as to the role of the enigmatic PqsE.

It is well written and the experiments are logically performed and the figures are clear. I only have a couple of minor comments:

- 1) It is interesting that in a pqsE mutant, pyocyanin is not made and yet the phz genes are expressed (line 142). Do you have an explanation for this? This could merit a couple of lines of discussion.
- 2) Line 210. Remove 'of' from the title
- 3) I had to tick the box of 'not satisfactory stats'. I appreciate that some of the differences look large between conditions, but some more specific stats tests when comparing key conditions would enhance the data.

We thank the editor and reviewers for their very constructive comments. In the provided annotated revised manuscript, changes are shown using the *Track changes* feature of Word. The lines where specific changes have been made are also specified below.

Reviewer comments:

Reviewer #1 (Comments for the Author):

In this manuscript the authors revisit the multilayered *Pseudomonas aeruginosa* quorum sensing (QS) circuitry, which includes the *las*, *rhl*, and *pqs* QS systems which involve the 3-oxo-C12-HSL, C4-HSL, and PQS signaling molecules, respectively. In this hierarchy the *las* QS system is first activated during growth followed by the downstream *rhl* and *pqs* systems. Recently, it has also become appreciated that LasR, and not RhlR, is the major driver of C4-HSL production in WT *P. aeruginosa*. However, it is becoming increasingly apparent that the *lasR* regulator is frequently mutated during chronic infections, and QS genes can be still be expressed in *lasR* mutants. Here the authors investigated the role of the C4-HSL regulator RhlR, the C4-HSL synthetase RhlI, and PqsE in modulating QS gene expression in *lasR* mutant backgrounds. Using a series of mutants generated in the PA14 reference strain, the authors measured QS gene expression, QS virulence factors, and C4-HSL signaling molecule production. Ultimately, these experiments show that PqsE enhances RhlR-dependent QS in a *lasR* mutant background.

Overall, this is an interesting paper that makes an important contribution towards our understanding of *lasR* mutant *P. aeruginosa*, which are common mutants found in many chronic infections, including cystic fibrosis. The experiments are thoroughly controlled and employ a number of complementary methods. However, the experimental design lacks statistical tests to compare the different mutants. The authors plot most data as QS gene expression vs culture density. This temporal analysis is valuable, because it shows that some genes are activated at later phases of growth in the different mutant backgrounds. Yet, in its current presentation, the data analyses lack statistical rigor, and these comparisons are not as strong as they might be if a statistical test was included for each experiment. While the data were not captured for all strains at exactly the same optical density, the data could be binned in a way to compare the different mutants at comparable ODs (see major comments below).

No new experiments are needed to back up the conclusions in this paper; however, reanalysis with statistics would strengthen the data.

Major comments

The abstract is somewhat confusing and could be written more clearly. In its current form, the abstract goes from background information to results, to background information and more results. It makes it somewhat challenging to determine what was known before this study, what was done (no methods are mentioned), and what was actually determined. Admittedly, this is a complicated story, but the abstract should be rewritten for clarity.

Author's response: We agree with this comment. As noted by the reviewer, this is challenging, especially considering that we already had reached the maximum length allowed (250 words), but we have done our best to rewrite the abstract for improved clarity and completeness, following the reviewer's suggestions

Figures: The line graphs could be maintained to still show the trends, but the data could be plotted a second way to enable multiple ANOVA tests to compare C4-HSL production and reporter activities at different phases of growth. For example, the data could be binned by OD600. In Figure 2 the bins might be OD600 0-1.5, 1.8-2.2, 2.5-3.5, 3.75-4.0, 4.1-4.5, 5.0-6.0. Within each bin, an ANOVA test could be performed with multiple comparisons to determine if strains are statistically significantly different in the given experimental measurements at the different phases of growth.

Author's response: One-way analyses of variance (ANOVA) with Tukey *post hoc* tests were performed at different stages of growth, as suggested by the reviewer. Our analyses confirm that all stated conclusions are based on significant differences between the strains. We now mention this in the material and method section (L357). Since there are many conditions (strains) on each graph, statistical significance is now only mentioned when appropriate throughout the manuscript. The whole output of our analyses is however available for consultation; do you think it should be included in the Supplemental material?

Some of the figures (1, 5, and 6) were blurry.

Author's response: Indeed. This was corrected, and the new version should have clear figures.

The authors do not discuss any of the limitations to their study. Some limitations should be addressed, including the use of only one strain, PA14; as well as the use of many marked mutants. Why weren't clean deletions used? Have the mutants used in

this study been previously complemented?

Author's response: We clarified that the study should be extended to more strains in the Conclusion section (L296-299).

All the mutants used in the paper have been previously used and complemented in other published studies (mentioned in Table 1), with the exception of *lasR::Gm/ΔpqsE* and *rhII::MrT7/ΔpqsE*, which we made for the present study. We chose to make them from the same single mutant backgrounds to limit strain-to-strain variations. All mutants are in genes which are not organized in operons, limiting any impact on adjacent genes.

Reviewer #2 (Comments for the Author):

This paper examines the roles of PqsE and RhIR specifically in the absence of LasR. Because *lasR* mutants are commonly found in infections, it is important to understand the QS hierarchy in the light of this and this paper sheds further welcome information as to the role of the enigmatic PqsE.

It is well written and the experiments are logically performed and the figures are clear. I only have a couple of minor comments:

1) It is interesting that in a *pqsE* mutant, pyocyanin is not made and yet the *phz* genes are expressed (line 142). Do you have an explanation for this? This could merit a couple of lines of discussion.

Author's response: We actually start seeing production of pyocyanin in cultures of the WT strain at $OD_{600} \cong 2.5$ even if the transcription of *phz* is activated before that (see Fig. 3 of the manuscript). The expression of the *phz* operons might be just too low to detect any pyocyanin. Also, since pyocyanin is not the direct product of the *phz* operons, it is possible another enzyme (e.g. PhzM or PhzS) implicated in the conversion of the precursor in pyocyanin does not follow the same pattern of expression. This has been added to the manuscript (L153-159)

2) Line 210. Remove 'of' from the title

Author's response: This has been corrected in the manuscript (L222).

3) I had to tick the box of 'not satisfactory stats'. I appreciate that some of the differences look large between conditions, but some more specific stats tests when comparing key conditions would enhance the data.

Author's response: One-way analyses of variance (ANOVA) with Tukey *post hoc* tests were performed at different stages of growth. Our analyses confirm that all stated conclusions are based on significant differences between the strains. This is now mentioned in the material and method section (L357). Statistical significance is now cited when comparing certain conditions in the text. The whole output of our analyses is available for consultation.

May 5, 2020

Dr. Eric Déziel
INRS-Armand-Frappier Santé Biotechnologie
531 Boul. des Prairies
LAVAL, QC H7V1B7
Canada

Re: mSystems00194-20R1 (PqsE is essential for RhIR-dependent quorum sensing regulation in *Pseudomonas aeruginosa*)

Dear Dr. Eric Déziel:

Your manuscript has been accepted, and I am forwarding it to the ASM Journals Department for publication. For your reference, ASM Journals' address is given below. Before it can be scheduled for publication, your manuscript will be checked by the mSystems senior production editor, Ellie Ghatineh, to make sure that all elements meet the technical requirements for publication. She will contact you if anything needs to be revised before copyediting and production can begin. Otherwise, you will be notified when your proofs are ready to be viewed.

Sincerely,

Elizabeth Shank
Editor, mSystems

Journals Department
Supplemental Material: Accept
Supplemental Material: Accept
Supplemental Material: Accept